# Microbial mineralization of cellulose in frozen soils

Javier H. Segura[1], Mats B. Nilsson[1], Mahsa Haei[1], Tobias Sparrman[2], Jyri-Pekka Mikkola[2,3], John Gräsvik[4], Jürgen Schleucher[5] & Mats G. Öquist[1]

High-latitude soils store ~40% of the global soil carbon and experience winters of up to 6 months or more. The winter soil $CO_2$ efflux importantly contributes to the annual $CO_2$ budget. Microorganisms can metabolize short chain carbon compounds in frozen soils. However, soil organic matter (SOM) is dominated by biopolymers, requiring exoenzymatic hydrolysis prior to mineralization. For winter SOM decomposition to have a substantial influence on soil carbon balances it is crucial whether or not biopolymers can be metabolized in frozen soils. We added $^{13}$C-labeled cellulose to frozen (−4 °C) mesocosms of boreal forest soil and followed its decomposition. Here we show that cellulose biopolymers are hydrolyzed under frozen conditions sustaining both $CO_2$ production and microbial growth contributing to slow, but persistent, SOM mineralization. Given the long periods with frozen soils at high latitudes these findings are essential for understanding the contribution from winter to the global carbon balance.

[1] Department of Forest Ecology & Management, Swedish University of Agricultural Sciences (SLU), Skogsmarksgränd, Umeå SE-901 83, Sweden. [2] Department of Chemistry, Umeå University, Umeå SE-901 87, Sweden. [3] Industrial Chemistry & Reaction Engineering, Process Chemistry Centre, Åbo Akademi University, Åbo-Turku FI-20500, Finland. [4] Iggesund Paperboard, Iggesund SE-825 80, Sweden. [5] Department of Medical Biochemistry and Biophysics, Umeå University, Umeå SE-901 87, Sweden. Correspondence and requests for materials should be addressed to J.H.S. (email: javier.segura@slu.se) or to M.G.Ö. (email: mats.oquist@slu.se)

High-latitude ecosystems store an estimated 40% of the Earth's soil carbon (C) pool of ca. 3000 Pg C[1]. This is >3.5 times the amount of present $CO_2$–C in the atmosphere[2]. Thus, the release of even a small proportion of the soil C pool can profoundly affect atmospheric $CO_2$ levels and global climate[2, 3]. During winter, which can last for up to 6 months, $CO_2$ losses from boreal forests soils can amount to ca. 20% of the annual C emitted[4–7]. This occurs despite low-soil temperatures and frozen conditions. Thus, winter emissions make an important contribution to the annual net C balance of seasonally frozen boreal soils.

Soil microbial populations can remain metabolically active and produce $CO_2$ also when soils are frozen. Substrate labeling studies have shown that microorganisms in frozen soils can use readily dissolved monomeric C compounds metabolically to sustain both catabolic and anabolic activity[8, 9]. However, soil organic matter (SOM) is dominated by large biopolymeric C forms[10] requiring exoenzymatic hydrolysis before they can be utilized metabolically and it is not known whether such processes occur in frozen soils. From a soil C balance perspective this is important; if microbial activity in frozen soils is restricted to using dissolved monomers already present in the soil solution when soils freeze, this activity would mainly influence the timing of mineralization, since these compounds will eventually be mineralized when the soil thaws. However, if there is both hydrolysis of biopolymers by exoenzymes and concomitant mineralization of the substrates formed, the winter season has a more profound impact on the soil C balance.

It has been suggested that complex SOM components may be inaccessible to microorganisms during winter because of direct effects of low temperature on exoenzymatic activity that may inhibit hydrolysis[11]. While high potential exoenzymatic activities have been found in boreal soils down to 4 °C[12], there is little information on the activity of exoenzymes in frozen boreal forest soils. The diffusion of exoenzymes in the frozen soil matrix may be restricted by low liquid water content inhibiting enzymatic access to the available substrate[13]. On the other hand, frozen soils typically contain an unfrozen liquid water pool[14]. For example, in boreal forest soils, the size of this unfrozen water pool has been directly linked to the SOM content[14]. Öquist et al.[15] reported liquid water contents of 0.40–0.65 g $H_2O$ g $SOM^{-1}$ at −4 °C in the organic horizons of boreal forest soils, representing pore size equivalents with unfrozen water of 0.08–0.14 μm[16]. In this pool of

unfrozen water, diffusion of substrates to and from microbial cells can be sustained[17]. Thus, it is not evident from available literature that exoenzymatic activity in frozen boreal forest soils must be unconditionally inhibited.

We hypothesized that soil microorganisms in boreal forest soils can hydrolyze, metabolize and grow on organic biopolymers under frozen conditions, and tested the hypothesis by analyzing the microbial utilization of $^{13}$C-labeled cellulose incubated with boreal forest soil samples at −4 °C for 195 days. Cellulose was chosen as a model substrate since carbohydrate biopolymers generally constitute 40–45% of the surface O-horizon in boreal forest soils[18] and cellulose is the most common biopolymer, typically comprising 20–30% of the plant litter mass[19]. A replicate set of soil samples was also incubated at 4 °C to validate microbial viability and to evaluate impacts of soil freezing on cellulose decomposition. The transformation rates of $^{13}$C-cellulose to $^{13}$C water-soluble carbohydrates (monomers and oligomers), $^{13}$C-$CO_2$, and the incorporation of $^{13}$C into membrane phospholipid fatty acids (PLFAs) were determined. We show that soil microbial communities can hydrolyze cellulose in the frozen soils and use the released substrate for both catabolic and anabolic metabolism.

## Results

**Microbial $CO_2$ production.** $CO_2$ was biogenically produced from the labeled cellulose and inherent SOM pool in the soil samples incubated at both 4 and −4 °C. At 4 °C there were stable rates of both total $CO_2$ production and $^{13}CO_2$ production during the initial 14 days, corresponding to 0.43 and 0.31 mg of $CO_2$ and $^{13}$C-$CO_2$ g SOM $day^{-1}$, respectively (Fig. 1a). After 14 days the production rates gradually declined towards the end of the incubation period. After 14 days, ca. 6.5% of the added $^{13}$C-labeled cellulose had been mineralized to $^{13}$C-$CO_2$, accounting for 68% of the total $CO_2$ production (Fig. 1a).

At −4 °C there were stable rates of both total $CO_2$ production, as well as $^{13}CO_2$ production during the initial 113 days, corresponding to 0.006 and 0.0005 mg of $CO_2$ and $^{13}$C-$CO_2$ g SOM $day^{-1}$, respectively (Fig. 1b). After 113 days the rates gradually declined towards the end of the incubation period. After 113 days, ca. 0.24% of the added $^{13}$C-labeled cellulose had been mineralized to $^{13}$C-$CO_2$, accounting for ca. 16% of the total $CO_2$ production (Fig. 1b). At both temperatures, no significant

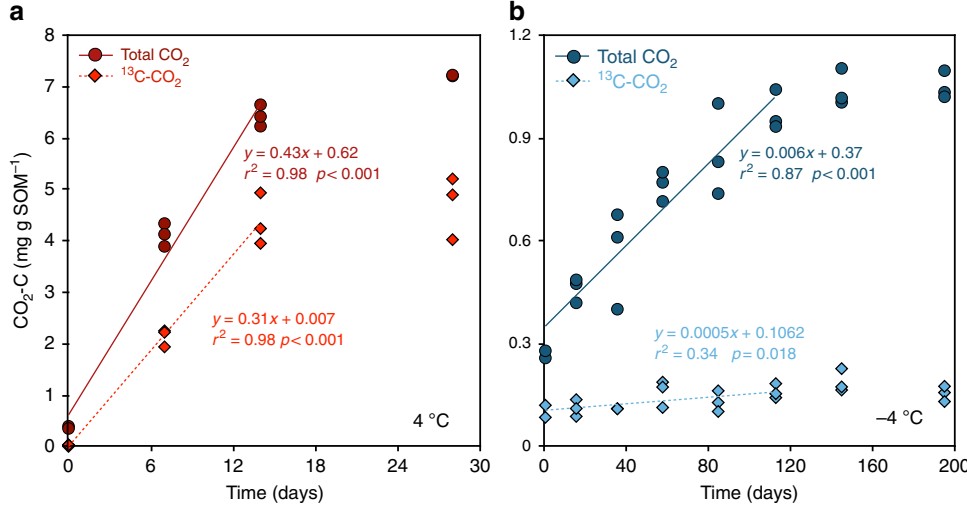

**Fig. 1** $CO_2$ and $^{13}$C-$CO_2$ from the added $^{13}$C-cellulose were produced in both unfrozen and frozen soil samples. Changes in $CO_2$ and $^{13}$C-$CO_2$ concentrations with time (showing fitted linear functions) in the samples incubated at 4 °C (over 14 days; panel **a**, red circles and diamonds) and −4 °C (over 113 days; panel **b**, blue circles and diamonds)

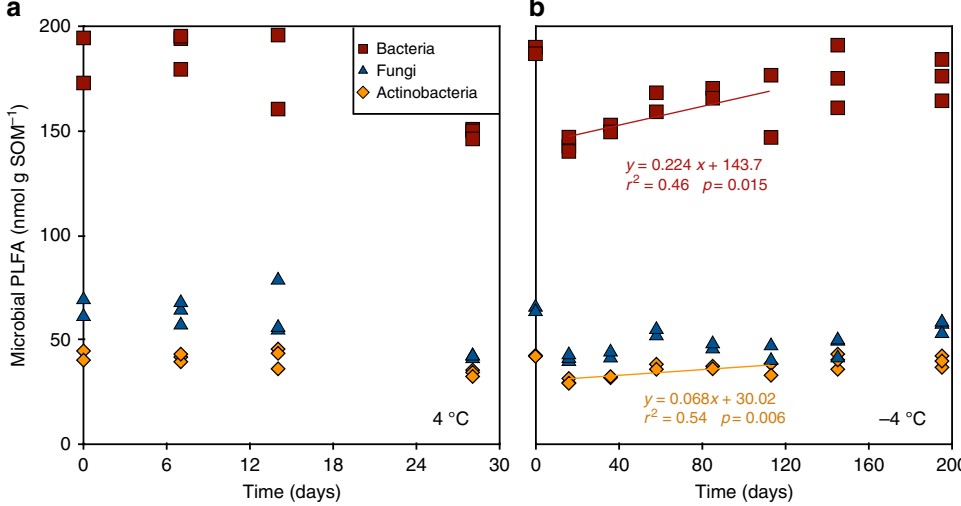

**Fig. 2** Concentration of bacterial and actinobacterial phospholipid fatty acid (PLFA) markers increased significantly over time in the frozen samples. Changes with time in bacterial (red squares), fungal (blue triangles) and actinobacterial (orange diamonds) PFLA concentrations in soils incubated at 4 °C (panel **a**) and −4 °C (panel **b**). Linear regressions in panel **b** represents significant increases in concentrations of bacterial and actinobacterial PLFA markers until day 113, during which $CO_2$ production occurred in the samples (cf. Fig. 1b)

$CO_2$ production was detected in control samples in which microbial activity was inhibited by sodium azide ($NaN_3$) (data not shown).

**Changes in microbial PLFA abundance.** In samples incubated at 4 °C, the concentrations of bacterial, fungal and actinobacterial PLFA markers did not change significantly during the incubation time (Fig. 2a). In samples incubated at −4 °C, the concentrations of bacterial, fungal and actinobacterial PLFA markers decreased by 25%, 35% and 29%, respectively, during the first 16 days. Thereafter, concentrations increased significantly and linearly over time (until day 113) for bacterial and actinobacterial ($r^2 = 0.46$, $p = 0.015$ and $r^2 = 0.54$, $p = 0.006$) but not for fungal PLFA markers (Fig. 2b).

**Synthesis of $^{13}$C-labeled PLFAs.** Newly synthesized $^{13}$C-enriched lipid chains were detected in the incubated soil samples at both temperatures (Fig. 3). The chemical shifts corresponded well with those of the polar lipid metabolites i.e. phospholipids, constituents of microbial cell membranes (Supplementary Fig. 2). At the two investigated temperatures, the region representing the aliphatic chain of the enriched lipid molecules was dominated by at least 90% (C2 and C4–CΩ-1 signal peaks) and 80% (C3 and CΩ signal peaks) of the expected lipid signals shifts. At −4 °C, the median enrichment between day 1 and day 195 corresponded to 41 and 26% for the C2 and C4–CΩ-1 signals, respectively. For the C3 and CΩ signals, the observed median enrichment is 41 and 36% (Kruskal–Wallis test, $p < 0.05$) (Fig. 3 and Supplementary Fig. 2).

**Microbial production of $^{13}$C-labeled WSC.** At 4 °C, the abundance of $^{13}$C-labeled water-soluble carbohydrates ($^{13}$C-WSC) did not significantly change in $NaN_3$ inhibited control samples ($p = 0.14$). In biologically active samples at 4 °C, the $^{13}$C-WSC concentration increased linearly by 0.08 mg $^{13}$C-WSC g SOM$^{-1}$ per day ($r^2 = 0.88$, $P < 0.001$, $n = 12$) from 0.7 mg $^{13}$C-WSC g SOM$^{-1}$ immediately after temperature equilibration to 3.02 mg $^{13}$C-WSC g SOM$^{-1}$ at day 28 (Fig. 4a).

At −4 °C, the abundance of $^{13}$C-WSC did not significantly change in metabolically inhibited control samples ($p = 0.58$). In biologically active samples at −4 °C, the $^{13}$C-WSC concentration increased linearly after temperature equilibration ($r^2 = 0.51$, $p = 0.004$, $n = 14$) at a rate of 0.008 mg $^{13}$C-WSC g SOM$^{-1}$

per day, peaking at 1.65 mg $^{13}$C-WSC g SOM$^{-1}$ at day 113 (Fig. 4b).

At 4 °C, the amount of $^{13}$C-WSC produced from the added $^{13}$C-labeled cellulose was positively correlated with the amount of $^{13}$C-$CO_2$ respired over time (Pearson $r = 0.85$. $p = 0.004$), but not to changes in PLFA markers for any particular microbial group. At −4 °C, the net increase in $^{13}$C-WSC released from the added $^{13}$C-cellulose was positively correlated to the amount of $^{13}$C-$CO_2$ respired until day 113 (Pearson $r = 0.66$, $p = 0.014$). After the initial decline in microbial PLFA concentrations (observed from day 0 to day 16), the net $^{13}$C-WSC concentration was positively correlated to the increases in bacterial and actinobacterial PLFA concentrations between days 16 and 113 (Pearson $r = 0.57$, $p = 0.05$ and Pearson $r = 0.66$, $p = 0.019$ respectively).

**Discussion**

Our results provide the first demonstration (to our knowledge) that microorganisms in frozen boreal forest soils can hydrolyze biopolymeric constituents of SOM and use the products both catabolically and anabolically. Despite the importance of biopolymer hydrolysis for soil C balances, previous studies on microbial activities under frozen conditions have focused on the use of labile C substrates[8, 9, 20–23] and their contribution to wintertime $CO_2$ fluxes[24, 25] However, the decomposition of C biopolymers is what regulates soil C balances and even small changes in the decomposition rates of the large biopolymeric SOM pool could over decades result in important changes in soil C stocks, and atmospheric $CO_2$ concentrations[13]. Seasonally frozen environments are common globally and the loss of carbon from frozen soils constitutes a positive radiative feedback missing in current coupled Earth-system model projections[2]. The observed hydrolysis and mineralization of cellulose in our frozen samples contradicts the prevailing view that freezing precludes biopolymer decomposition[11].

Immediately after adding labeled cellulose, $^{13}$C-WSC was detected in $NaN_3$ amended samples, thus confirming the presence of inherently active cellulases[26]. However, it was also evident that the observed cellulose hydrolysis is tightly coupled to an active microbial population, since significant net hydrolysis only occurred in frozen samples where microorganisms remained viable. Thus, the inherent enzymatic capacity originally present in

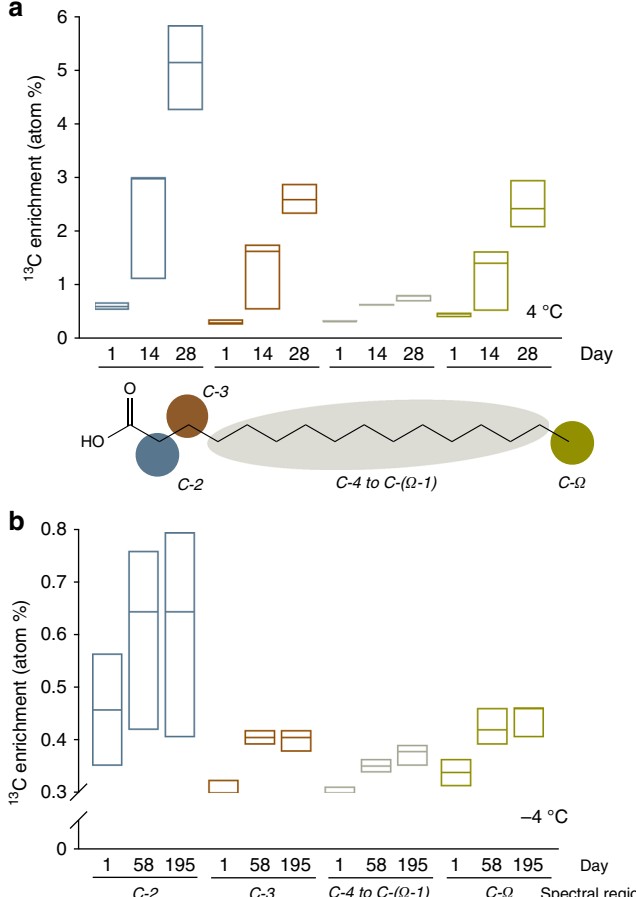

**Fig. 3** [13]C from the added [13]C-cellulose was used to synthetize new cell membrane lipids in both unfrozen and frozen soil samples. Incorporation of the [13]C label into cell membrane lipids as determined by 1D [1]H analysis and a 1D variant of a [1]H, [13]C Heteronuclear Single Quantum Coherence NMR spectroscopy, in incubations at 4 °C (panel **a**) and −4 °C (panel **b**). The bars show the median contents and max and min ranges of [13]C-enrichments (relative to natural abundance) in acyl chains at the start (day 1), mid-point (days 14 and 58 at 4 and −4 °C, respectively) and end (days 28 and 195 at 4 and −4 °C, respectively) of the incubations. We obtained signals for spectral regions assigned to the following hydrogen and corresponding C atoms (illustrated in the model phospholipid fatty acid chain): 2.28 p.p.m.-H-2 (blue circle and bars); 1.62 p.p.m.-H-3 (brown circle and bars); 1.28 p.p.m.-H-4 to H-(Ω-1) (gray elipse and bars) and 0.9 p.p.m.-H-Ω (green circle and bars). Differences in levels of [13]C-labeling between the start and the end of the incubation were examined with a Kruskal–Wallis test, which does not assume normality of data

the soil samples is lost when soils freeze, either because of the direct influence of temperature on enzyme stability[27] or an indirect effect on enzyme mobility due to reduced diffusion rates in the frozen matrix[13]. However, it is evident that the microbial activity at −4 °C led to the production of new functional enzymes capable of cellulose hydrolysis in the frozen soil.

The decline in PLFA concentrations during the first 2 weeks of incubation is in accordance with previous findings[28, 29] and suggests an initial destruction of microbial cells in response to freezing. The magnitude of this decline may have been influenced by the rather rapid drop in temperature when the samples were placed at −4 °C, meaning that the temperature change probably proceeded faster than it would have done under field conditions. However, it is evident that a large fraction of the microbial cells remained viable and that the microbial population was able to

grow at −4 °C because PLFA concentrations increased significantly from day 16 onwards. The increases in concentrations of bacterial and actinobacterial PLFA markers coincided with significant net increases in [13]C-WSC concentrations in our frozen samples, indicating that growing microorganisms actively synthesized exoenzymes to an extent that resulted in a net increase in WSCs. Both bacteria and actinobacteria are important decomposer communities in forest soils[30, 31] and our findings suggest this is also the case in frozen soils. Substrate available to bacteria and actinobacteria in our mesocosms may have become limited with time, probably as a result of diffusion constraints in the frozen soil[32], and is a likely explanation for the decline in activity after ca 3 months. Further, the increase in microbial PLFA concentrations correlated with total $CO_2$ production, suggesting that microbial carbon mineralization was closely related to changes in microbial biomass throughout the incubations. By the end of the incubations (after 195 days), total PLFA concentrations had returned to initial levels. Thus, microbial communities recover from initial freezing-induced destruction (or adapted communities may develop) even under constantly frozen conditions.

In addition to the microbial growth inferred from increases in PLFA concentrations during the incubations, the incorporation of [13]C from [13]C-cellulose into the PLFA pool confirms that the biopolymeric substrate was a source of C assimilated by the microbial populations in the frozen soil matrix. We detected homogeneous enrichment throughout the newly synthesized acyl chains in samples incubated at −4 °C (Fig. 3b, Supplementary Fig. 2), which are largely changed through elongation and branching (forming additional terminal, C- Ω, methyl groups)[33]. Localized enrichment also occurred in molecular fragments, particularly C-2 and C4–CΩ-1 groups of the acyl chains (Fig. 3b, Supplementary Fig. 2). This is consistent with observations that microbial growth at low temperatures affects the degree of unsaturation, chain length, and branching at the methyl end of fatty acids[33–35]. Thus, our observations indicate that adaptive changes permitting maintenance of metabolic processes occurred in the microbes' membranes under the frozen conditions.

The enrichment observed in PLFAs in the frozen samples amounted to 0.1–0.2 at%, while the labeled substrate accounted for ca. 16% of total respired $CO_2$. Assuming [13]C-$CO_2$ production made a similar contribution to PLFA synthesis, the total fraction of newly synthesized PLFAs as a result of [13]C-cellulose hydrolysis and concomitant assimilation would be around 1–2%. This may very well be an underestimation of the potential assimilation given the rapid freezing of the samples that did not give the microbial population the same possibility to adapt to low temperature as they would have had under natural winter progression. Understanding of the contribution from winter and frozen conditions to the soil carbon balance is essential given the long winters in high-latitude ecosystems. In addition, accounting for all contributions to SOM decomposition, and especially the biopolymeric constituents, is crucial.

Cellulose hydrolysis rates were slower in frozen soil than in unfrozen soil, probably largely due to the reduction in unfrozen water content restricting diffusion through the soil matrix. However, Tilston et al.[17] found an exponential relationship between unfrozen water content and temperature, with an abrupt change in microbial respiration rates at an unfrozen water fraction of ca. 13% (v/v) in soils sampled at the same site. For soil water to remain unfrozen at temperatures from −2 to −4 °C, water potentials of −4.8 to −2.4 MPa are required[16]. Thus, considerable physico-chemical changes associated with liquid water content and water potentials can arise from relatively small temperature increases in the frozen matrix, and major differences in unfrozen water content between different soil types and ecosystems can be expected[36–38].

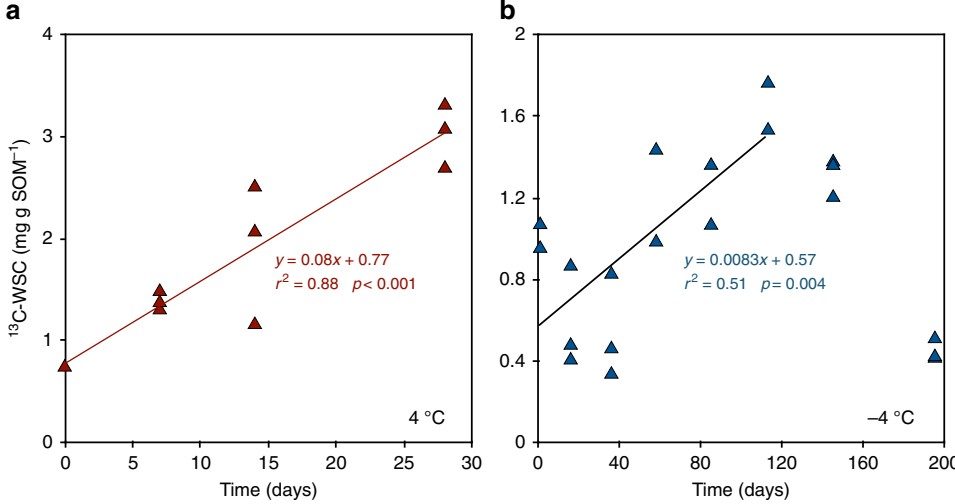

**Fig. 4** $^{13}$C-labeled water-soluble carbohydrates were form from hydrolysis of the added $^{13}$C-cellulose in both unfrozen and frozen soil samples. Changes with time (and fitted linear functions, $p < 0.005$) in concentrations of $^{13}$C-labeled water-soluble carbohydrates ($^{13}$C-WSC) in the samples incubated at 4 °C (panel **a**, red triangles) and −4 °C (panel **b**, blue triangles)

For such reasons, soil freezing has complex effects on numerous factors influencing metabolic activities, as shown by the differences in metabolic dynamics between our frozen and unfrozen samples. The behavior of unfrozen samples was generally consistent with previous findings[12, 18, 39]. It should also be noted that the amorphous cellulose structure and both N and P availabilities in our incubations were set to target optimal conditions for microbial activity. As for the cellulose structure, naturally occurring cellulose polymers from Norway spruce and Scots pine (the dominating tree species at the site sampled) contain around 50% amorphous cellulose[40], making it a common constituent of SOM. N and P were added to remove nutrient limitations to examine the role of, e.g., temperature and matrix related effects (e.g., unfrozen water content) specifically targeting the potential for cellulose depolymerization and mineralization in the frozen samples. It has been suggested that cellulose hydrolysis is hampered in nitrogen-limited soils[41, 42]. However, it has also been documented that the role of, e.g., substrate and nutrient availability on microbial activity becomes less important when soils freeze[17]. When soils freeze most of the soil pore water transitions into ice and the dissolved compunds becomes more concentrated in the remaining liquid water pool[16], at the same time as metabolic rates and nutrient demands are decreased[17].

Our findings extend the understanding of the microbial capacity to hydrolyze a major biopolymeric SOM component in seasonally frozen boreal soils, thus contributing to slow, but persistent, degradation of SOM. Questions remain as to whether this capacity is also applicable to the decomposition of other C biopolymers as well as over a wider range of environmental conditions. Nonetheless, because of the large contribution of biopolymers to soil C stocks, future investigations should include efforts to quantify this capacity and elucidate its contribution to C cycling in high-latitude ecosystems.

## Methods

**Soil sampling**. Cores (15 cm diameter) were collected in the middle of October 2013 from the organic (O)-horizon (0–5 cm) of typical boreal Spodosols[43] at a site dominated by *Picea abies* L. and *Pinus sylvestris*, L. in the Kulbäcksliden Experimental Area, northern Sweden (64°11′N, 19°33′E), with understory and field vegetation dominated by *Vaccinium myrtillus* L., *V. vitis idaea* L., and *Pleurozium schreberi* (Brid.) Mitt. See[15, 18] for further site details. Organic surface layers were used because they contain most of the soil carbohydrate biopolymers[18] and are particularly affected by soil frost[44]. Winter soil temperatures of −5 to −1 °C are a common temperature range in boreal snow covered systems[45] and the incubation

temperature of −4 °C represents the typical winter soil temperature regime in the area from which the soil samples were collected[44]. On collection, litter, moss and underlying mineral soil were removed from the samples, which were pooled into a large single composite sample to maximize representativeness. The soils were transported directly to the laboratory where the composite was homogenized by passing it through a sieve (6 × 3.5 mm mesh) in its field-moist state. Needles, coarse and visible fine roots and other debris were removed manually. The homogenized soil was then stored at −20 °C[46] for not > 2 months until the start of incubations. We have previously found that this storage time does not significantly affect microbial activity after thawing[17]. Subsamples were used to determine the dry weight (24 h at 105 °C) and SOM content (by loss on ignition; 6 h at 550 °C). The water content was 2.43 g $H_2O$ g dw$^{-1}$, organic matter content 0.955 g g$^{-1}$ with C and N contents of 0.524 ± 0.003 (SD) and 0.14 ± 0.001 g g$^{-1}$, respectively and C:N mass ratio 37.4 ± 2.6 ($n = 10$) (Elemental Analyzer, Flash EA 2000, Thermo Fisher Scientific, Bremen, Germany[47]).

**Preparation of amorphous $^{13}$C-cellulose**. $^{13}$C-cellulose (97 at% $^{13}$C, from *Zea mays* L., obtained from Iso*Life*, Wageningen, The Netherlands) was treated with ionic liquid (IL) to disrupt its partly crystalline structures and increase its accessibility to microorganisms[48, 49]. This also removes impurities (e.g., hemicelluloses, xylose, proteins and organic acids) found in the labeled product. We used the IL 1-butyl-3-methylimidazolium chloride (BmimCl[50]) because the mild temperature treatment (~ 75 °C, 48 h) does not create toxic compounds (e.g., sugar degradation products) or degradation of cellulose to glucose (cellulose treatments with chloride base ILs at 120 °C for 24 h do not exhibit significant amounts of cellulose degradation[51]).

All starting materials used to synthesize the IL BmimCl were purified before use. 1-Methylimidazole (Sigma-Aldrich 99%) was distilled, in a vacuum, from potassium hydroxide (Akzo Nobel 87.9%). 1-Chlorobutane (Acros Organics 99%) was distilled from CaO (Sigma-Aldrich). Ethyl acetate (Fisher chemicals HPLS) and acetonitrile (Fisher chemicals HPLS) were dried by passage through neutral alumina oxide 90 (Merck 70–230 mech ASTM). All purified starting materials were used within 4 h of purification. All reactions and purification steps were carried out under dry argon using Schlenk techniques.

A volume of 89 ml of 1-methylimidazole (1.1 mol) was dissolved in ~100 ml ethyl acetate at room temperature, 116 ml of 1-chlorobutane (1.1 mol) was added drop-wise to the mixture over the course of an hour, and the reaction mixture was refluxed at 75 °C for 48 h. The reaction vessel was then allowed to cool to room temperature before being carefully sealed and incubated at −20 °C for 16 h. The upper organic layer was then removed by Schlenk filtration. A volume of 100 ml of ethyl acetate was added to the sludge-like bottom IL layer and thoroughly shaken. At this point the IL precipitated out as white crystals that were washed again with 50 ml ethyl acetate. The crystals were then dissolved in ~50 ml of warm acetonitrile and recrystallized overnight at −20 °C. The upper organic layer was again removed and the crystals were washed three times with 50 ml of ethyl acetate before being stored under a strong vacuum overnight to remove residual solvent.

The dissolution of cellulose in BmimCl was carried out in four batches. In total, 6.7 g of $^{13}$C-cellulose (Iso*Life*, Wageningen) was mixed with 235 g BmimCl in a nitrogen atmosphere. For every batch, the mixture was then stirred vigorously with 400 ml of methanol (Sigma-Aldrich) at ~ 70 °C, after which the precipitated cellulose was separated, washed and vacuum-filtered with methanol (3×100 ml) and subsequently with deionized water (5×100 ml). Because Cross Polarization-

Magic Angle Spinning Nuclear Magnetic Resonance (CP-MAS NMR) spectroscopy indicated that the resulting product retained methanol, it was washed 100 times with deionized water to achieve total removal of the methanol. Finally, the resulting gel was homogenized in a blender and vacuum-filtered to yield a product containing 93% (w/w) water. The IL-induced changes in the cellulose structure were evaluated by CP-MAS NMR and a Flash EA 2000 Elemental Analyzer (Thermo Fisher Scientific, Bremen, Germany), which revealed complete conversion of the polymer into an amorphous form and estimated 92.3 at% $^{13}C$ content (Supplementary Fig. 1). This confirmed the removal of organic impurities (e.g., hemicellulose, xylose, proteins and organic acids) that could otherwise have confounded the results.

**Soil incubations**. After thawing at 4 °C for 12 h, 2.234 g (wet weight) sub-samples of the soil, with ca. 0.9 g dw SOM, were placed in autoclaved 60 ml glass serum bottles. A 2.110 g portion of wet gel of the resulting $^{13}C$-labeled amorphous cellulose was added to each bottle and manually mixed with the soil, thereby adding ca. 67 mg $^{13}C$ g$^{-1}$ SOM. Thus, the added C amounted to ca. 15% of the organic C in each incubated sample. A volume of 70 μl of a solution containing $(NH_4)_2$ SO$_4$ and KH$_2$PO$_4$ was then added, to give a final C:N:P molar ratio of 182:13:1, which is reportedly optimal for microbial growth in similar soil samples[52-54]. Addition of nutrients was undertaken to remove the influence of factors other than temperature and liquid water that can influence our assessment of the potential for enzymatic hydrolysis of the labeled cellulose. To measure any abiotic transformation of the $^{13}C$ label, samples with inhibited microbial activity were prepared by adding 450 μl NaN$_3$ solution (77 mM) to replicate bottles (four for incubation at each test temperature) immediately after adding cellulose gel and nutrient solution to the soil and just before incubation[9]. The bottles were sealed with butyl rubber septa, evacuated, refilled with atmospheric air (~400 p.p.m. CO$_2$) then placed in temperature-controlled cabinets. The background CO$_2$ and $^{13}CO_2$ concentrations were corrected for when calculating production rates. The whole sample preparation procedure took <1 h. The bottles ($n = 24$) were incubated at −4 °C for 195 days. To check for and ensure microbial viability in the soil samples, a replicate set of soil samples was also incubated at 4 °C ($n = 18$) for 28 days. This also allowed us to compare rates of hydrolysis and activity above and below the freezing point to evaluate impacts of soil freezing on cellulose decomposition. On each sampling occasion (weekly at 4 °C, monthly at −4 °C), 450 μl of NaN$_3$ solution was added to each of three bottles to inhibit further microbial activity. Headspace gas samples were withdrawn from these bottles and transferred to N$_2$-flushed GC vials to determine the total CO$_2$ content using a gas chromatograph (Perkin-Elmer Auto Systems, Waltham, MA, USA) equipped with a methanizer and flame ionization detector[55], and to N$_2$-flushed EXETAINER® tubes containing 500 μl 0.5 M KOH to determine their $^{13}C$-CO$_2$ contents, as described below.

**Determination of $^{13}C$-CO$_2$**. Amounts of $^{13}C$-CO$_2$ respired during the incubations were determined by $^{13}C$ NMR analyses of CO$_2$ absorbed by the KOH solutions in the EXETAINER® tubes[39]. After injecting the headspace gas, each tube was equilibrated at 4 °C for 1 h, then 250 μl of the solution it contained and 250 μl of 1.0 M KCH$_3$COO were transferred to a NMR tube (Wilmad-Lab Glass, Vineland, USA) and analyzed using a 600 MHz Avance III HD spectrometer (Bruker Biospin GmbH, Rheinstetten, Germany), equipped with a 5 mm Broad Band Observe Cryo-Probe. The acetate carbonyl signal at 181.4 p.p.m. was used as a natural abundance internal reference to integrate the signal of the $^{13}C$ carbonate at 168.2 p.p.m. To obtain quantitative $^{13}C$ spectra for these carbonyl carbons, the transmitter frequency and relaxation delay were set to 175 p.p.m. and 300 s (>5 $T_1$), respectively, and the acetate methyl group was $^1H$ decoupled at 1.8 p.p.m. using mild WALTZ-16 decoupling during acquisition.

**Determination of $^{13}C$-labeled water-soluble carbohydrates**. To monitor changes in contents of $^{13}C$ water-soluble carbohydrate monomers and oligomers ($^{13}C$-WSC), incubated soil samples were ground and homogenized in a liquid N$_2$ bath (SPEX Freezer/Mill 6850, Metuchen, USA). From each homogenized sample, 2.6 g (wet weight) was transferred to a 50 ml Falcon tube (Sarstedt AG&Co, Nümbrecht, Germany), 5 ml of water was added per g of soil, the sample was shaken at 200 r.p.m. for 1 h then centrifuged at 3000 r.p.m. for 15 min[56]. The supernatant was passed through a 0.45 μm filter and freeze-dried, while the residual solid phase was freeze-dried and kept at −20 °C until further extraction for PLFA analysis (see below). The freeze-dried supernatant was then rewetted and dissolved in one ml of water containing 10% D$_2$O and vortex-mixed for one min. A sample of 500 μl of the resulting mix was transferred to a NMR tube (Wilmad-Lab Glass) and vortex-mixed for another minute. Next, $^{13}C$ spectra of the samples were recorded using the 600 MHz spectrometer and probe mentioned above, with a 30° $^{13}C$ excitation pulse, $^1H$ WALTZ-16 decoupling during 0.9 s acquisition and a 2 s relaxation delay. All $^{13}C$-WSC spectra were dominated by $^{13}C$- $^{13}C$ coupled C, presumably originating from the added $^{13}C$-cellulose, and the detected natural $^{13}C$ was insignificant relative to signals from the added $^{13}C$-cellulose.

**Analysis of membrane phospholipid fatty acids**. To determine their phospholipid fatty acid (PLFA) concentrations, 0.5 g portions of the freeze-dried and homogenized soil samples were extracted and fractionated following Bligh & Dyer

method, with modifications by Frostegård et al.[57]. The abundance of PLFAs was quantified using a Perkin-Elmer Clarus 500 gas chromatograph (Waltham, MA, USA). PLFA quantification yields results comparable to other biomass related methods and has been proposed as a suitable method for detecting increases in biomass after, for example, the addition of labeled substrates[58]. In total 30 FAs were detected and identified, and the PLFAs i15:0, a15:0, i16:0, 16:1ω9, 16:1ω7t, 16:1ω7c, i17:0, a17:0, cy17:0, 18:1ω7, cy19:0 were used as bacterial markers[59, 60], 18:2ω6,9 as a fungal marker[60, 61] and 10Me16, 10me17 and 10me18 as actino-bacterial markers[59, 62, 63].

To analyze $^{13}C$-label incorporation into PLFAs, 5 mg portions of lipids extracted as above were weighed into smaller glass vials, evaporated at 40 °C under a stream of N$_2$ and re-weighed. Following addition of 400 μl methanol-d$_4$ (99.8 at%, Cambridge Isotope laboratories, Andover, MA) and 200 μl chloroform (99.8 at%, Armar Chemicals, Döttingen, Switzerland) to the extracts, 480 μl portions of the resulting solutions were transferred to NMR tubes (Wilmad-Lab Glass) and examined by means of a one-dimensional (1D)$^1H$ analysis and a 1D variant of a $^1H$, $^{13}C$ Heteronuclear Single Quantum Coherence (HSQC) analysis[64-67] using a 500 MHz Avance III spectrometer (Bruker Biospin GmbH). The latter analysis provided information on protons connected by a single chemical bond with $^{13}C$ carbons. The signal of the residual CD$_2$HOD solvent has a certain intensity ratio between the 1D $^1H$ and the $^1D$ HSQC spectra, which is determined by the natural $^{13}C$ abundance of methanol. This ratio was then evaluated for PLFA signals: increases in the ratio, compared with that found for the CD$_2$HOD signal, revealed the level of enrichment in cell membranes of soil microorganisms.

The 1D HSQC pulse sequence used in our study consisted of three stages: First, a relaxation delay to restore $^1H$ equilibrium polarization followed by polarization transfer from $^1H$ to $^{13}C$ and back to $^1H$ using two Insensitive Nuclei Enhanced by Polarization Transfer (INEPT) steps to obtain a $^1H$ signal arising only from protons directly bound to $^{13}C$[68, 69] and finally, the acquisition of a $^1H$ signal under $^{13}C$ decoupling. We identified and evaluated four potential sources of uncertainty with the method that could have influenced our results. The first potential source of uncertainty is that $^1H$ must be fully, or at least equally, relaxed to represent spin populations and to avoid incomplete $T_1$ relaxation between scans. Thus, the relaxation delay must be long enough to allow for full $^1H$ longitudinal (spin-lattice) $T_1$ relaxation. This was tested with saturation recovery experiments in which the slowest relaxing signal, the residual $^1H$ from CD$_3$OD, was fully relaxed to within 1% after 20 s (we tested delays of 5, 10, 20, 60, 90 and 240 s). Thus, all signals were over 99% relaxed with the 26 s relaxation delay used in the study.

The second potential source of uncertainty is that there could be a differential signal loss during the HSQC sequence due to $T_2$ relaxation, as well as variation of J couplings. The $^1H$ HSQC involves two INEPT steps with at total J-evolution delay of $4 \times 1/(4 \times J_{HC}) = 1/J_{HC}$ or about 7.1 ms with our chosen coupling constant $J_{HC} = 140$ Hz. During the J-evolution, the $^1H$ signal decays with transversal (spin–spin) $T_2$ relaxation. The total J-evolution delay should thus ideally be short compared with the $T_2$ relaxation time or alternatively the $T_2$ relaxation times should be as similar as possible, so that variation in $T_2$ does not bias quantification. In our case the longest $T_2$ and sharpest line is expected for the residual $^1H$ signal CHD$_2$OD of CD$_3$OD. The full width at half height for the individual lines in the CHD$_2$OD was below 1 Hz for all samples, indicating a $T_2$ of longer than 0.3 s. The PLFA line shapes are complex and overlap makes it hard to estimate individual linewidths. From the saturation recovery experiment, it is clear that all lipid CH$_2$ signals were relaxed at 5 s, but the terminal CH$_3$ needed up to 10 s, indicating a $T_1$ of about 1 s for CH$_2$ and 2 s for CH$_3$. For small molecules in solution $T_2 \approx T_1$ so it is likely that $T_2$ for individual lines is of this order. Assuming a $T_2$ or effective $T_2^*$, as low as 0.2 s would still lead to a maximum decay of 4% during the total J-evolution of 7.1 ms. However, as already mentioned, it is rather the difference in $T_2$ between the lipid and the reference methanol signal that is relevant, and it is thus more likely that this will lead to differences in $T_2$ relaxation of <2%. The J-coupling delay was selected to optimize the transfer of methanol signal, but also works well for the smaller J couplings in the lipid chain (3% drop in efficiency for methyls). The $^{13}C$ transmitter was set to 40 p.p.m., i.e., approximately centered between the methanol $^{13}C$ signal at 48.5 p.p.m. and the lipid chain signals at 35 to 15 p.p.m. Thus, the differential $T_2$ relaxation during the HSQC and J coupling variation can together cause an error in relative signal intensities of only ~5%, entirely sufficient to quantify $^{13}C$ enrichment. The third potential source of uncertainty involves the comparability of line shapes between $^1H$ 1D and 1D HSQC with $^{13}C$ decoupling. To get comparable line shapes between the 1D $^1H$ and the 1D HSQC experiments, both were processed using 2048 points in the frequency domain and 5 Hz line broadening. For this comparison, it is important that the $^{13}C$ decoupling during $^1H$ acquisition is sufficient to yield similar line shapes from the 1D $^1H$ and 1D HSQC experiments. To achieve this, we used 3.2 kHz WALTZ decoupling centered at 40 p.p.m. For the signals of interest (methanol and lipids) the line shapes obtained are identical for the two experiments. Thus, the comparability of line shapes in the 1D $^1H$ and 1D HSQC spectra was assured by strong $^{13}C$ decoupling and identical processing.

The fourth potential source of uncertainty is the influence of spectral overlap in the acyl region which we addressed by running a 2D HSQC experiment (Supplementary Fig. 2). Although the acyl region obviously contains overlapping signals, the 2D HSQC signals are totally dominated by lipid chain signals ($^1H$, $^{13}C$) with C2 at (2.28 p.p.m., 34.1 p.p.m.), C3 (1.62 p.p.m., 24.3 p.p.m.), C4–CΩ-1 (1.28 p·p.m., main $^{13}C$ peak at 29–30 p.p.m., but with CΩ-2 at 32.8 p.p.m and CΩ-1 at

23.4 p.p.m.) and finally CΩ at (0.90 p.p.m., 14.0 p.p.m.). The 2D HSQC signals with most overlap in the 1D $^1H$ are C3 and CΩ, where about 80% of the acyl signals are centered on expected peaks (1.62 p.p.m., 24.3 p.p.m.) and (0.90 p.p.m., 14.0 p.p.m.), respectively. The C2 and C4–CΩ-1 peaks are dominated (> 90%) by the expected lipid signals shifts. This should be compared with the observed median increase in enrichment between day 1 and day 195 in our frozen samples, which for C2 and C4–CΩ-1 correspond to 41% and 26%, respectively; much greater than the < 10% overlap. For C3 and CΩ, with a potential overlap of < 20%, the observed median enrichment is 41 and 36%. Thus, the observed enrichment in the PLFAs goes beyond the potential influence of spectral overlap in the acyl region (Supplementary Fig. 2).

**Statistical analysis.** The Prism package (Graph Pad Software 6.0, La Jolla, USA) was used for all statistical analyses. Differences in $^{13}C$ incorporation into PLFAs were examined using the Kruskal Wallis-test, which requires the measurements to be placed in rank-order but does not assume normality of data. Differences were regarded as significant if $P < 0.05$.

**Data availability.** The solid-state and liquid-state NMR data are available at Figshare, DOI: 10.6084/m9.figshare.5318932. All other relevant data supporting the findings of this study are available within the article, the Supplementary Information or upon request from the authors.

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

## Acknowledgements

We acknowledge financial support from the Swedish National Research Council (contract 621-2011-4874), the Swedish Research Council Formas (contract 214-2013-834), The Kempe Foundation, (contract JCK1107), The Knut and Alice Wallenberg Foundation (2011.0228), the Carl Tryggers Foundation (contract 13:536). We also acknowledge the NMR for Life facilities at Umeå University, the Bio4Energy program and the Wallenberg Wood Science Center.

## Author contributions

M.G.Ö. secured funding and developed the idea with M.B.N. and J.S. J.H.S. and M.H. carried out field work and conducted laboratory incubations together with M.G.Ö, and NMR analysis together with T.S. and J.S. J.G., M.H. and J-P.M. carried out the Ionic Liquid preparation and cellulose treatment. J.H.S. carried out statistical analysis with input from M.G.Ö., M.B.N., T.S. and J.S. J.H.S. wrote the first draft of the manuscript and all authors commented on manuscript drafts and contributed to writing.

## Additional information

**Competing interests:** The authors declare no competing financial interests.

