## [Peer Review File · Nature Communications]

Reviewers' Comments:

Reviewer #1 (Remarks to the Author)

This manuscript is a detailed and clearly written exploration of the mineralization of cellulose in boreal organic-horizon soil communities near freezing. The authors very cleverly explore both microbial catabolism and anabolism of cellulose (via NMR) and associated changes in the microbial community, albeit at a coarse scale (PLFAs). However, it does not appear that the findings are a significant advancement in the understanding of how microbial communities process carbon below freezing.

Although an interesting and valuable study, SIP in soils incubated below freezing is not entirely novel (e.g. McMahon, Wallenstein, Schimel 2009, Tuorto et al 2014, Rivkina et al 2000, Hunger et al 2011), and it is not clear that the findings would be of broad interest, especially given the limited scope of the discussion. The discussion could place the key findings within the context of what they mean for microbially mediated C-cycling in boreal forests and associated model predictions of C flux. One of the major strengths of this paper is the analysis of ¹³C-labelled cellulose and degradation products, which seems to be a novel contribution to this field of study. The methods are generally clearly described, although clarification is required on a few points which are detailed below.

Line 23: this statement is not entirely true, please revise to fully characterize the composition of SOC.

Line 25: this sentence is missing a word

Line 38: what exactly do the authors imply by "substantial"? Please clarify.

Line 55: this depends on the depth of freeze, snow cover, and soil type; in Alaskan boreal forest soils surface soils (≤ 10 cm) regularly freeze below -5 degrees C for most of the winter (≥ 3 months).

Line 76: when were the soils collected? what was the depth of organic layer soils collected?

Line 88: how long were the soils stored at -20 oC between collection and analysis? Under what conditions were the soils frozen and thawed? What it an immediate freeze/thaw or was the temperature brought down slowly (as it would be under natural conditions)? This should be indicated and discussed.

Line 103: Why did the authors add fertilizer to the treatments? This could have potentially major effects on microbial community and should be discussed.

Line 136: Here and elsewhere, spell numbers less than 10 in full, unless in a series that includes at least one number >10 .

Line 173: how was the normality of data tested? Many p-values are marginally significant (e.g. line 198). Without addressing how the data was assessed for normality prior to running statistics with normality assumptions, it is hard to know if the significance numbers reported here are technically correct.

Line 314: authors should also consult Frostegard et al 2011 ("Use and misuse of PLFA measurements in soils", SBB).

Line 326: the authors are missing quite a few key studies which explore carbon use by microbial communities at or near freezing.

Line 329: this statement requires a citation and to be more fully developed.

Reviewer #2 (Remarks to the Author)

This paper describes a study in which mineralisation of cellulose in frozen and unfrozen organic soils was investigated using ¹³C-labelled cellulose. While it is a potentially useful contribution to the area, I feel the authors have over-interpreted (and at times mis-interpreted) their data and that, as a result, the manuscript is not suitable for publication without substantial revision.

Major points:

1. It is not clear to me whether or not the NMR method used to determine the degree of enrichment in PLFAs is novel. No reference is provided for the method (e.g. around lines 158-171) suggesting its novelty. If this is the case, then I think it is necessary to demonstrate its reliability in accurately measuring the degree of ^{13}C enrichment in less complex samples. The basis of the technique is only rudimentally described as comparing the signal intensity ratio between signals in 1D ^1H spectra and 2D HSQC spectra for residual solvent and peaks assigned to PLFAs. I can imagine several potential complicating factors, including signal overlap, differences in peak shape between samples, differences in relaxation rates, and before I'd be willing to accept the values derived from this approach, I'd need to be convinced that these factors have been considered and accounted for. If this is an established procedure, references to its development and validation need to be provided. Quite a bit of reliance is placed on this line of evidence, so it needs to be convincing.
2. The choice of curve in the fits to the data in Figure 1 are poor. What is the basis for choosing a logarithmic function? I can think of no reason to expect CO_2 production to follow such a trajectory and the derived fitted parameters have no physical meaning. And why would you not fix the intercept at zero, when by definition there is zero CO_2 produced at the start of the incubation? The shape of the curve is clearly wrong in panel (a) – a high r^2 for a fit to a non-linear function with just four time points provides no confidence in its credibility – clearly the fit is poor at 7 days. A much better case could be made for an initial linear phase from 0-14 days at 4°C and 0-100 days at -4°C , with CO_2 production rates slowing after that.
3. The fits and their interpretation for Figure 2 are not much better. Again, what is the justification for a logarithmic function in panel (b)? Also, the y-axes are very confusing.
4. The interpretation of Figure 4b as having two phases is not valid. "Phase II" is entirely predicated on the one data point at ~ 190 days, while "phase I" seems strongly reliant on the point at ~ 120 days and no reason is given for ignoring the zero point as being part of "phase I". In any case, why would you have expected two linear phases with a changeover at 120 days?
5. Figure 5 is similarly overly contrived. By choosing these specific time periods (which don't correspond with those of Figure 4 or any other suggested "phases" identified in the paper) it appears the authors are "cherry-picking" data points to get statistical significance. Even then it is marginal at best and unconvincing.
6. At many points the discussion is too speculative, in part because it is based on what I consider invalid over-interpretation of the data. This undermines the strength of the study, which is to clearly demonstrate microbial processes are occurring at -4°C and to give an indication of the rates relative to rates in unfrozen soil at 4°C . By trying to push the data too far, the authors fail to make the convincing case they could have.

Reviewer #3 (Remarks to the Author)

General

The authors tested the decomposition of ^{13}C -labeled cellulose in the organic layer of a boreal forest soil at $+4^\circ\text{C}$ and at -4°C . They show that mineralization of soil organic matter in general and of the added cellulose in particular happens at both temperatures, i.e., also below the freezing point, but that mineralization rates are much lower at -4°C than at $+4^\circ\text{C}$, and that cellulose decomposition is especially reduced. The authors conclude that in frozen soil, microorganisms can not only take up and metabolize small organic compounds, but also decompose larger polymers that represent the bulk of the soil organic matter, albeit at low rates.

While the findings are novel and the experimental support seems sound, the motivation for this study and the implications are not clear. Why is it important if microorganisms can decompose polymeric organic matter in frozen soil, and what do the findings mean for the functioning of high latitude ecosystems? I suggest to extend introduction and discussion accordingly.

See below for specific questions and comments.

Abstract

Line 23: Change "solely" to "mostly" or similar.

Introduction

I find the writing generally okay, but the introduction – especially the first two paragraphs – would profit from language polishing.

Line 41: "occurs"

Line 52: Change to "and overall carbon use efficiency is lower"

Lines 57-59: I suggest changing the sentence to "While exoenzymes are active in these soils down to 0°C, with temperature response (Q10) values slightly lower than two, there is little information on exoenzyme activity in frozen soils." (or similar)

Line 63: Re-place the comma by a full stop.

Material & Methods

Line 80: Remove ")" after "Mitt."

Line 90: How were C and N determined?

Lines 95-102: Did you check if the ionic liquid treatment left residuals in the cellulose (e.g., toxic compounds, nutrients) that could affect microbial activity? Why did the at% change during cellulose purification, and how did you determine the new value?

Line 110: Did the added air contain CO₂, and if yes, at what concentration?

Line 145: Change "a further minute" to "another minute".

Line 154: What method/instrument was used for PLFA quantification?

Line 157: Change "actinomycete" to "actinobacteria" throughout the manuscript.

Using NMR for measuring ¹³C in CO₂ and PLFAs (instead of IRMS) is not very common in ecology. What were your reasons for choosing this approach?

Results

Line 197: What time points did you compare to test for significance?

Line 218: Delete "here".

Lines 245-246: Delete "linked to any particular microbial group".

Line 248: Are the data not shown in Figure 2b?

Discussion

Line 263: Delete "glucose moieties" (I found it confusing).

Lines 266-269: I cannot find support for this statement in Biasi et al. (2001), and only very vaguely in Wallenstein et al. (2009). The reference Grogan et al. (2001) is missing in the reference list. Please check all your references carefully.

Lines 277-280: Conant et al. (2008) writes about the temperature sensitivity of decomposition, without speculating about temperature sensitivities of different enzymes.

Line 301: Do you mean enrichment in CO₂?

Line 302: I think "microbial activity" is more suitable than "microbial growth rates".

Lines 314-316: I cannot follow this sentence. What do you mean with "temporal partitioning in utilization of the polymer substrate by different microbial groups"?

Lines 323-326: None of the three cited references shows an increasing importance of fungi from early to late winter! Please check your references.

Lines 327-329: The concentration of WSC is determined by the balance between production and consumption. A decrease in WSC could result from a decrease in cellulose breakdown, but also from an increased uptake of WSC by microorganisms (at constant, reduced or increased cellulose breakdown rates).

Lines 329-336: This is very speculative, and I would like to see it phrased more carefully.

Figures

Figure 2. I do not find it necessary to plot the different markers on different scales (Figure 2b). The figure would be easier to read if all were on the same scale. Where the logarithmic functions significant?

Figure 3. Change "cell membrane lipids" to "phospholipid fatty acids" in the legend, unless the NMR method was not specific for PLFAs.

Figure 5. Change the x axes labels to "bacterial and actinobacterial PLFAs" (Figure 5a) and "fungal PLFAs" (Figure 5b).

Reviewer #1

This manuscript is a detailed and clearly written exploration of the mineralization of cellulose in boreal organic-horizon soil communities near freezing. The authors very cleverly explore both microbial catabolism and anabolism of cellulose (via NMR) and associated changes in the microbial community, albeit at a coarse scale (PLFAs). One of the major strengths of this paper is the analysis of ^{13}C -labelled cellulose and degradation products, which seems to be a novel contribution to this field of study. The methods are generally clearly described, although clarification is required on a few points which are detailed below.

We appreciate that reviewer #1 identifies the analysis of ^{13}C -labelled cellulose and degradation products to be a novel contribution to this field of study.

Main comments

1). It does not appear that the findings are a significant advancement in the understanding of how microbial communities process carbon below freezing. Although an interesting and valuable study, SIP in soils incubated below freezing is not entirely novel (e.g. McMahon, Wallenstein, Schimel 2009, Tuorto et al 2014, Rivkina et al 2000, Hunger et al 2011), and it is not clear that the findings would be of broad interest, especially given the limited scope of the discussion

Response: The novel aspects of this study does not involve the application of stable isotope probing *per se* for investigations of microbial activity as a mediator of soil carbon mineralization. SIP for this purpose has been used for decades (e.g. by the current co-authors) and more recently specifically for frozen soils systems (as indicated by the exemplified references mentioned by the reviewer as well as others that we cited in the originally submitted manuscript). However, what is novel is the use ^{13}C SIP for investigating the hydrolysis of biopolymeric constituents of soil organic matter in a frozen soil matrix and the concomitant microbial utilization of the formed products.

To date, the general view conveyed in the scientific literature suggests that hydrolysis of biopolymeric SOM constituents in soils does not occur under frozen conditions due to the combination of direct low temperature effects on exo-enzymatic activity and the physical constrains on diffusion due to low liquid water content induced by freezing of soil water. However, these speculations are based on inconclusive data and have, to our knowledge, not been tested scientifically.

If the documented carbon mineralization in frozen soils would solely rely on simple monomers already present in the soil when it freezes this activity would be restricted to influence only the timing of mineralization and have very little impact on soil carbon balances. I.e. if these monomers are not mineralized in the frozen soil, they will be readily mineralized after spring thaw. However, if also exo-enzymatic hydrolysis can proceed in the frozen matrix we are dealing with processes that can have much more profound implications for the overall soil carbon balance of seasonally frozen systems.

It is evident from the reviewer #1s comment that we have not adequately explained in sufficient detail how our work contributes to the understanding of soil carbon dynamics below freezing.

Action: Further elaboration on the novelty, rationale, as well as implications, of our study have been made in the introduction (lines 34-55) and the discussion (lines 148-161 and 251-258).

2). The discussion could place the key findings within the context of what they mean for microbially mediated C-cycling in boreal forests and associated model predictions of C flux.

Action: Further elaboration on the implications of our findings in the context of C dynamics in boreal forest added in the discussion (lines 148-161 and 251-258).

Line comments

Line 23: this statement is not entirely true, please revise to fully characterize the composition of SOC.

Action: Changed text in line 22 to *“However, soil organic matter (SOM) is dominated by biopolymers, requiring exo-enzymatic hydrolysis prior to mineralization”*

Line 25: this sentence is missing a word

Action: The sentence has been revised now in line 23

Line 38: what exactly do the authors imply by “substantial”? Please clarify.

Action: Changed text in line 37 to *“During winter, which can last for up to six months, CO₂ losses from boreal forests soils can amount to ca. 20 % of the annual C emitted⁴⁻⁷. This occurs despite low soil temperatures and frozen conditions. Thus, winter emissions make an important contribution to the annual net C balance of seasonally frozen boreal soils.”*

Line 55: this depends on the depth of freeze, snow cover, and soil type; in Alaskan boreal forest soils surface soils (≤ 10 cm) regularly freeze below -5 degrees C for most of the winter (≥ 3 months).

Action: To motivate our choice of incubation temperature we have changed text in line 268 to *“Winter soil temperatures of -5 to -1 °C is a common temperature range in boreal snow covered systems⁴⁵ and the incubation temperature of -4 °C represents the typical winter soil temperature regime in the area from which the soil samples were collected⁴⁴”*.

Line 76: when were the soils collected? what was the depth of organic layer soils collected?

Action: Information added in the methods (lines 261-266)

Line 88: how long were the soils stored at -20 °C between collection and analysis? Under what conditions were the soils frozen and thawed? What it an immediate freeze/thaw or was the temperature brought down slowly (as it would be under natural conditions)? This should be indicated and discussed.

Action: Information was added in the methods (lines 276-278) and discussion lines (174-196).

Line 103: Why did the authors add fertilizer to the treatments? This could have potentially major effects on microbial community and should be discussed.

Response: The aim of the study was to investigate the potential for enzymatic hydrolysis of the labelled cellulose and concomitant microbial utilization of the formed substrates in a frozen soil matrix. To achieve this, nutrients were added to remove the influence of factors other than temperature and liquid water that could otherwise have influenced results.

Action: We have added information in the methods (Lines 299-301) and discussion (lines 237-249) motivating our experimental design and the rationale for adding nutrients to the samples and how it influences interpretations.

Line 136: Here and elsewhere, spell numbers less than 10 in full, unless in a series that includes at least one number >10.

Action: Changed accordingly.

Line 173: how was the normality of data tested? Many p-values are marginally significant (e.g. line 198). Without addressing how the data was assessed for normality prior to running statistics with normality assumptions, it is hard to know if the significance numbers reported here are technically correct.

Response: After the validation our approach to assess the level of ^{13}C enrichment in PLFAs, we used Kruskal Wallis statistics to test differences in the median incorporation of the labelled substrate between the start and end of the incubation time. The test assumes the measurements on a rank-order scale but does not assume normality of data.

Action: We have revised the statistical analysis and added information about this in lines (392-396) in the methods.

Line 314: authors should also consult Frostegard et al 2011 (“Use and misuse of PLFA measurements in soils”, SBB).

Action: We agree with the reviewer and have included this paper in the manuscript in methods, lines 357-359.: *“PLFA quantification yields results comparable to other biomass related methods and has been proposed as a suitable method for detecting increases in biomass after, for example, the addition of labelled substrates⁵⁵”*

Line 326: the authors are missing quite a few key studies which explore carbon use by microbial communities at or near freezing.

Action: We have revised our reference list and included evident studies to give appropriate credit to previous work in this area.

Line 329: this statement requires a citation and to be more fully developed.

Action: With the revisions made in accordance with reviewer #2's comments (see below) this statement is no longer valid.

Reviewer #2:

This paper describes a study in which mineralisation of cellulose in frozen and unfrozen organic soils was investigated using ^{13}C -labelled cellulose. While it is a potentially useful contribution to the area, I feel the authors have over-interpreted (and at times mis-interpreted) their data and that, as a result, the manuscript is not suitable for publication without substantial revision.

Major points:

1. It is not clear to me whether or not the NMR method used to determine the degree of enrichment in PLFAs is novel. No reference is provided for the method (e.g. around lines 158-171) suggesting its novelty. If this is the case, then I think it is necessary to demonstrate its reliability in accurately measuring the degree of ^{13}C enrichment in less complex samples. The basis of the technique is only rudimentally described as comparing the signal intensity ratio between signals in 1D ^1H spectra and 2D HSQC spectra for residual solvent and peaks assigned to PLFAs. I can imagine several potential complicating factors, including signal overlap, differences in peak shape between samples, differences in relaxation rates, and before I'd be willing to accept the values derived from this approach, I'd need to be convinced that these factors have been considered and accounted for. If this is an established procedure, references to its development and validation need to be provided.

Quite a bit of reliance is placed on this line of evidence, so it needs to be convincing.

Response: This is a most valid concern. The quantification of HSQC per se is not new. It has been generally used as a tool in metabolic flux analysis of cell cultures, but also for global analysis of cell wall metabolites (i.e. Szyperski, 1995; Mahrous *et al.*, 2008). A high precision (%) in quantification can be achieved as demonstrated by Martineau *et al.*, 2013. 2D HSQC has been previously used to assess the metabolic capacity of soil microorganisms (e.g. Fan *et al.*, 2009) and also to quantify low-percent ^{13}C enrichment in $^{13}\text{CO}_2$ labelling experiment on poplar, in a quite similar application to our study (Mahboubi *et al.*, 2015). The 1D variant of HSQC has also been used for quantification in metabolic flux analysis (e.g. Hansen & McCormack, 2002). Nevertheless, we agree that the specific approach used to quantify ^{13}C enrichment of PLFAs using 1D HSQC should be addressed in more detail as well as scrutinized and validated.

The 1D HSQC pulse sequence consist of three different stages: 1) relaxation delay to restore ^1H equilibrium polarization, 2) polarization transfer from ^1H to ^{13}C and back to ^1H using two Inensitive Nuclei Enhanced by Polarization Transfer (INEPT) steps to obtain ^1H signal arising only for protons directly bond to ^{13}C (Morris, 1980; Bax & Morris, 1981) and 3) the acquisition of ^1H signal under ^{13}C decoupling. From this setting we can identify three potential sources of problems with the method, as well as a fourth related to the reviewer's concern of potential signal overlap:

1) Incomplete T_1 relaxation between scans

2) Differential signal loss during the HSQC sequence due to T_2 relaxation, as well as variation of

J couplings

3) Comparability of lineshapes between ^1H 1D and 1D HSQC with ^{13}C decoupling

4) Signal overlap.

To resolve these concerns, we have re-analyzed our results as well as carried out additional NMR experiments and from this we can conclude that (detailed estimations in small font):

1) Measurement of T_1 relaxation times shows that all signals are >99% relaxed.

^1H must be fully, or at least equally, relaxed to represent spin populations. Thus the relaxation delay must be long enough to allow for full ^1H longitudinal (spin-lattice) T_1 relaxation. This was tested with saturation recovery experiments in which the slowest relaxing signal, the residual ^1H from CD_3OD , was fully relaxed to within 1% after 20s (we tested with 5, 10, 20, 60, 90 and 240 s relaxation delay). Thus, all signals were over 99% relaxed with the 26s relaxation delay used in the study.

2) Differential T_2 relaxation during the HSQC and J coupling variation can together cause only an approximately 5% error in relative signal intensities, fully sufficient to quantify ^{13}C enrichment.

The ^1H HSQC involves two INEPT steps with a total J-evolution delay of $4 \cdot 1/(4 \cdot J_{\text{HC}}) = 1/J_{\text{HC}}$ or about 7.1 ms with our chosen coupling constant $J_{\text{HC}} = 140\text{Hz}$. During the J-evolution the ^1H signal decays with transversal (spin-spin) T_2 relaxation. The total J-evolution delay should thus ideally be short compared to the T_2 relaxation time or alternatively the T_2 relaxation times should be as similar as possible so that variation in T_2 does not bias quantification. In our case the longest T_2 and sharpest line is expected for the residual ^1H signal CHD_2OD of CD_3OD . The full width at half height for the individual lines in the CHD_2OD was below 1Hz for all samples indicating a T_2 of longer than 0.3s. The PLFA line shapes are complex and overlap makes it hard to estimate individual line widths. From the saturation recovery experiment it is clear that all lipid CH_2 signals were relaxed at 5 s, but the terminal CH_3 needed up to 10s indicating T_1 of about 1 s for CH_2 and 2 s for CH_3 . For small molecules in solution $T_2 \approx T_1$ so it is likely that T_2 for individual lines are of this order. Assuming a T_2 or effective T_2^* as low as 0.2s would still lead to maximum decay of 4% during the total J-evolution of 7.1 ms. However, as already mentioned it is rather the difference in T_2 between the lipid and the reference methanol signal that is relevant, and it is thus more likely that this will lead to differences in T_2 relaxation of less than 2%. The J-coupling delay was selected to optimize the transfer of methanol signal, but also works well for the smaller J couplings in the lipid chain (3% drop in efficiency for methyls). The ^{13}C transmitter was set to 40 ppm, i.e. approximately centered between the methanol ^{13}C signal at 48.5ppm and the lipid chain signals at 35 to 15 ppm. Altogether this yields a good coverage of the spectral region of interest and the HSQC step will be accurate to about 5% with our settings.

3) Comparability of line shapes in 1D ^1H and 1D HSQC spectra has been assured by strong ^{13}C decoupling and identical processing.

To get comparable line shapes between the 1D ^1H and the 1D HSQC experiments, both were processed using 2048 points in the frequency domain and 5 Hz line broadening. For this comparison it is important that the ^{13}C decoupling during ^1H acquisition is sufficient to yield similar line shapes in the 1D ^1H and 1D HSQC experiment. To achieve this, we used 3.2 kHz WALTZ decoupling centered at 40 ppm. For the signals of interest (methanol and lipids) the obtained line shapes are identical for the two experiments.

4) 2D NMR shows that the signals of the 1D HSQC that were used for quantification are composed to 80 - >90% of lipid signals. The observed enrichment in the PLFAs goes beyond the potential influence of spectral overlap in the acyl region.

To address the question of spectral overlap in the acyl region, we ran a 2D HSCQ (see Supplementary Figure 2). Although the acyl region obviously contains overlapped signals, the 2D HSQC signals are totally dominated by lipid chain signals (^1H , ^{13}C) with C2 at (2.28ppm, 34.1ppm), C3 (1.62ppm, 24.3ppm), C4 - C Ω -1 (1.28ppm, main ^{13}C peak at 29-30ppm, but with C Ω -2 at 32.8ppm and C Ω -1 at 23.4ppm) and finally C Ω at (0.90ppm, 14.0ppm). Looking at the 2D HSQC the signals with most overlap in the 1D ^1H are C3 and C Ω where about 80% of the acyl signals are centered at expected peaks (1.62 ppm, 24.3ppm) and (0.90ppm, 14.0ppm), respectively. The C2 and C4 - C Ω -1 peaks are dominated to more than 90% by the expected lipid signals shifts. This should be compared to the observed median increase in enrichment between day 1 and day 195 in our frozen samples, which for C2 and C4 - C Ω -1 corresponds to 41% and 26%, respectively; much greater than the <10% overlap. For C3 and C Ω with a potential overlap of <20% the observed median enrichment is 41% and 36%.

New figure added to SI (Supplementary Figure 2): 2D HSQC spectrum of a representative soil lipid extract in chloroform/methanol overlaid with the corresponding 1D ^1H spectrum. The colored stripes indicate the ^1H projections interpreted as lipid chain signals C2, C3, C4 - C Ω -1 and C Ω , respectively. The residual ^1H signal due to deuterated methanol solvent is also indicated. This particular sample also contains ethylmethylketone as internal standard, with e.g. the methyl peak at (2.15ppm, 28.3ppm). No overlap is

visible for the C2 signal and the C4 - C Ω -1 strip is also totally dominated by the lipid signal (see asterisk Supplementary Figure 2 and text above). For C3 and C Ω the signal overlap corresponds to ca. 20%.

In conclusion, problems 1-3 above yield a quantification problem of potentially up to 5%. Spectral overlap might lead to uncertainties on the level of about 10-20% for phospholipid content of the analysed ^1H chemical shifts. However, even if the assignments are not 100% unique the signals still show a high ^{13}C enrichment in the acyl region dominated by lipid chains.

Action: 1. More information explaining the set-up of the NMR experiment (along the lines given above) has been inserted in the revised manuscript, methods (lines 364-390) and additional information relating to evaluation of the method is given in the supplementary information document (including Supplementary Figure 2 in response to comments).

2. The issue of potential signal overlap is addressed in the revised manuscript, methods (lines 388-390) and Supplementary information document (lines 129-143) in relation to data treatment. We now broaden our statistical evaluation of ^{13}C PLFA enrichment to test the whole acyl chain and from this we can conclude that we have a ^{13}C enrichment under the incubation time evidencing PLFA synthesis by the organisms in the frozen soil matrix.

3. Based on the potential overlap in some of the specific regions we have removed the part of the results and discussion related to the site specific enrichments at the four different positions of the PLFA chain, as well as the comparison between frozen and unfrozen conditions in this context.

2. The choice of curve in the fits to the data in Figure 1 are poor. What is the basis for choosing a logarithmic function? I can think of no reason to expect CO₂ production to follow such a trajectory and the derived fitted parameters have no physical meaning. And why would you not fix the intercept at zero, when by definition there is zero CO₂ produced at the start of the incubation? The shape of the curve is clearly wrong in panel (a) – a high r² for a fit to a non-linear function with just four time points provides no confidence in its credibility – clearly the fit is poor at 7 days. A much better case could be made for an initial linear phase from 0-14 days at 4°C and 0-100 days at -4°C, with CO₂ production rates slowing after that.

Response: We acknowledge this concern and we have revised the data interpretation and Figure 1 accordingly. We now identify an initial linear phase of CO₂ production from 0-14 days at 4°C and from 0-113 days at -4°C. Linear regressions are used to evidence significant buildup of CO₂ over time and to calculate CO₂ production rates. The same approach has been used for the $^{13}\text{CO}_2$ production. We have, however, kept the data points after this phase in the figure to clearly show how the rates tend to level-off at the end of the incubations.

Action: We have plotted new figures to accommodate the linear regressions and added information about it in the results in lines (88-145)

3. The fits and their interpretation for Figure 2 are not much better. Again, what is the justification for a logarithmic function in panel (b)? Also, the y-axes are very confusing.

Action: The function in Fig 2b has been changed to a linear function adopting the same scheme as applied for the CO₂ data in Fig 1 with a linear increase up to 113 days. In this regression we

have, however, excluded the time point at zero days, since it is evident that the amounts of PLFAs drop during the initial part of the incubation. We motivate this with the fact that this kind of initial microbial response to freezing has been documented in the literature (e.g. Feng *et al.*, 2007; Schmitt *et al.*, 2008). This figure shows that during the time phase with linear increase in $\text{CO}_2/^{13}\text{CO}_2$ (up to 113 days), we have a concomitant significant increase in bacterial and actinobacterial PLFAs (but not fungal PLFAs). We have also removed the 2nd y-axis in panel b to enhance clarity according to the reviewer's request (which also was a revision suggested by reviewer # 3).

4. The interpretation of Figure 4b as having two phases is not valid. "Phase II" is entirely predicated on the one data point at ~190 days, while "phase I" seems strongly reliant on the point at ~120 days and no reason is given for ignoring the zero point as being part of "phase I". In any case, why would you have expected two linear phases with a changeover at 120 days?

Response: The "phase" concept applied in evaluating this figure stems from the observations of CO_2 production and PLFA increases during the incubations, which are linear up to 113 days and thereafter level off. In this study we have three independent measurements related to microbial activity at identical time resolution: CO_2 production, PLFA production, and production of ^{13}C -WSCs from ^{13}C -cellulose by enzymatic activity. Although the *measurements* are independent of each other, the *processes* are not. Thus, we find it justified to evaluate the WSC data in Figure 4b in the time window up to 113 days, since it coincides with the period where we have a linear increase in both $\text{CO}_2/^{13}\text{CO}_2$ and PLFAs. We do, however, acknowledge the reviewer's concern of omitting the zero point and have included it in the revised regression. It should also be noted that the ^{13}C -WSCs amounts we observe are net values resulting from the balance between production and consumption (cf. comment by reviewer 3 below), which probably to a large degree can explain the large variations observed. However, the manifestation of a significant net increase in WSCs over time provides evidence that there is an active hydrolysis of the added ^{13}C -cellulose in the frozen soil matrix, which coincides with both significant CO_2 production and significant PLFA synthesis.

As for the identified "phase II" in figure 4b we acknowledge the reviewer's concern and agree that the evaluation of this borders on over-interpretation of data given the few data points available after 113 days, and this has been deleted from the manuscript. However, we believe this by no means impacts the main findings and conclusion of the study that the microbial population can hydrolyze and grow on the added polymer under frozen conditions. Based on the reviewer's comments we also acknowledge that we have not been clear enough to motivate the rationales used for our data evaluation and interpretation.

Action: In addition to the revisions of the figure 4b outlined above we have added information in the results (lines 125-145) and discussion (lines 163-172)

5. Figure 5 is similarly overly contrived. By choosing these specific time periods (which don't correspond with those of Figure 4 or any other suggested "phases" identified in the paper) it appears the authors are "cherry-picking" data points to get statistical significance. Even then it is marginal at best and unconvincing.

The “phases” forming the basis for this figure were indeed derived from the concepts presented earlier in the manuscript and correspond to these time points, and thus, we find the reviewer’s notion of cherry-picking data erroneous. It is evident, however, that we have not conveyed this sufficiently. Figure 5b, relating to the end of the incubations (days 113-195) is deleted based on the same reasoning as outlined for the CO₂, PLFA and WSC data above. After careful consideration (and also taking into account reviewer #2 concern in point # 6 below) we have decided to remove also figure 5a from the manuscript. The important information in this figure is the correlation between WSCs and PLFAs demonstrating the significant link between these two processes during the first 113 days of incubation. However, the data per se is already presented in figures 2 and 3 (albeit individually and as functions of time), and in this context we judge figure 5a to be redundant. Instead, the key information, which is the significant correlation (R and p-values) between the independent measurements of microbial activity is mentioned in the main text (lines 125-145).

6. At many points the discussion is too speculative, in part because it is based on what I consider invalid over-interpretation of the data. This undermines the strength of the study, which is to clearly demonstrate microbial processes are occurring at -4°C and to give an indication of the rates relative to rates in unfrozen soil at 4°C. By trying to push the data too far, the authors fail to make the convincing case they could have.

Response and action: This concern is dealt with in the substantial revisions in data presentation and evaluation and a substantial revision of the discussion in response to reviewer #2 previous comments (see above and discussion lines 148-249). We judge that reviewer #2 concerns of invalid and over-interpretation of data is dealt with by the revisions outlined above. We acknowledge that addressing these concerns has greatly enhanced the focus of the paper which is to demonstrate the microbial capacity to use natural biopolymers for anabolic and catabolic processes in frozen soils.

Reviewer #3 (Remarks to the Author):

General

The authors tested the decomposition of ¹³C-labeled cellulose in the organic layer of a boreal forest soil at +4°C and at -4°C. They show that mineralization of soil organic matter in general and of the added cellulose in particular happens at both temperatures, i.e., also below the freezing point, but that mineralization rates are much lower at -4°C than at + 4°C, and that cellulose decomposition is especially reduced. The authors conclude that in frozen soil, microorganisms can not only take up and metabolize small organic compounds, but also decompose larger polymers that represent the bulk of the soil organic matter, albeit at low rates.

Main content

While the findings are novel and the experimental support seems sound, the motivation for this study and the implications are not clear. Why is it important if microorganisms can decompose

polymeric organic matter in frozen soil, and what do the findings mean for the functioning of high latitude ecosystems? I suggest to extend introduction and discussion accordingly. See below for specific questions and comments.

Action: The manuscript has been revised to better convey the novelty, rationale and implications of the study in the introduction (lines 34-55) and discussion (lines 148-161 and 251-258).

Abstract

Line 23: Change “solely” to “mostly” or similar.

Action: Changed accordingly.

Introduction

I find the writing generally okay, but the introduction – especially the first two paragraphs – would profit from language polishing.

Action: These paragraphs have been revised to accommodate this concern

Line 41: “occurs”

Action: Changed accordingly.

Line 52: Change to “and overall carbon use efficiency is lower”

Action: Changed accordingly.

Lines 57-59: I suggest changing the sentence to “While exoenzymes are active in these soils down to 0°C, with temperature response (Q₁₀) values slightly lower than two, there is little information on exoenzyme activity in frozen soils.” (or similar)

Action: Changed in lines (58-60).

Line 63: Re-place the comma by a full stop.

Action: Changed accordingly.

Material & methods

Line 80: Remove “)” after “Mitt.”.

Action: Changed accordingly.

Line 90: How were C and N determined?

Response: We used an elemental analyzer - Isotope Ratio Mass Spectrometer (EA-IRMS) (following the protocol by Werner *et al.*, 1999)

Action: We added information on the measurement principle and instrumentation in the methods (lines 278-283).

Lines 95-102: Did you check if the ionic liquid treatment left residuals in the cellulose (e.g., toxic compounds, nutrients) that could affect microbial activity? Why did the at% change during cellulose purification, and how did you determine the new value?

Response: When regenerating the cellulose from the ionic liquid (IL) solution, extensive washing with methanol and water was carried out (e.g. washed 100 times with deionized water in order to remove any traces of methanol). Analysis of CP-MAS NMR spectra showed a complete conversion of the polymer into an amorphous form and it also confirmed the removal of organic impurities. Moreover, we use the IL BmimCl (Ab Rani *et al.*, 2011) following a mild temperature treatment (75 °C, 48 h) and thus no toxic compounds (e.g. sugar degradation products etc.) or degradation of cellulose to glucose occurs (cellulose treatments with chloride base ILs at 120 °C for 24 h does not show significant amounts of cellulose degradation e.g.(Clough *et al.*, 2015).

The percentage ¹³C enrichment changes due to the fact that the IL treated cellulose represents the pure cellulose content whereas the original raw material contained hemicellulose, xylose, proteins and organic acids adding to the atomic percentage. The value for the new atomic percentage was determined using an elemental analyzer - Isotope Ratio Mass Spectrometer (EA-IRMS) (Werner *et al.*, 1999).

Action: Additional information on the IL treatment of cellulose and its effects has been added in the Supplementary Information (lines 3-52).

Line 110: Did the added air contain CO₂, and if yes, at what concentration?

Response: The air added was atmospheric air and thus contained ca. 400 ppm CO₂. This information has been added in methods (lines 305-307)

Line 145: Change “a further minute” to “another minute”.

Action: Changed accordingly.

Line 154: What method/instrument was used for PLFA quantification?

Response: We assessed the microbial community PLFA by using the method of Bligh and Dyer (1959) as modified by Frostegård *et al.* (1991). The abundance of PLFAs was quantified using a Perkin–Elmer Clarus 500 gas chromatograph (Waltham, MA, USA).

Action: We further elaborated on about PLFA methods and instrumentation in the methods section (lines 355-363).

Line 157: Change “actinomycete” to “actinobacteria” throughout the manuscript.

Action: Changed accordingly.

Using NMR for measuring ¹³C in CO₂ and PLFAs (instead of IRMS) is not very common in ecology. What were your reasons for choosing this approach?

Response: We acknowledge that NMR spectroscopy is not the most common tool used in ecology or by soil biogeochemists. We use NMR spectroscopy based techniques because it allows us to track the metabolic processes occurring in the frozen soil, without having to destroy

the sample structure. In many cases this means that we can use NMR samples in later, destructive, experiments. Since we use uniformly labelled substrates, preparation of samples and processing in NMR is faster, easier and less expensive compared to IRMS. Using NMR also allow us to pinpoint the specific molecular position of labelled substrates incorporated in microorganisms. Altogether, we can generally extract more information from NMR as compared to IRMS.

Results

Line 197: What time points did you compare to test for significance?

Response: We have revised the data interpretation and Figure 2. We now identify a linear phase (see Figure 2b) that confirmas a significant change over time. Thus, we do no longer evaluate significant differences between specific time points.

Line 218: Delete “here”.

Action: Changed accordingly.

Lines 245-246: Delete “linked to any particular microbial group”.

Action: Changed accordingly.

Line 248: Are the data not shown in Figure 2b?

Response and action: Originally, data was not shown in Figure 2a. We have revised the data interpretation and Figure 2 (see Figure 2a and 2b)

Discussion

Line 263: Delete “glucose moieties” (I found it confusing).

Action: Changed accordingly.

Lines 266-269: I cannot find support for this statement in Biasi et al. (2001), and only very vaguely in Wallenstein et al. (2009). The reference Grogan et al. (2001) is missing in the reference list. Please check all your references carefully.

Action:

- 1). With the revisions made this sentence has been deleted from the manuscript.
- 2). We have carefully checked the reference list to include all cited articles in the bibliography.

Lines 277-280: Conant et al. (2008) writes about the temperature sensitivity of decomposition, without speculating about temperature sensitivities of different enzymes.

Response and action: We agree with the reviewer and changed text in the discussion in lines (163-172) to “Immediately after adding labelled cellulose, ^{13}C -WSC was detected in NaN_3 amended samples, thus confirming the presence of inherently active cellulases²⁶. However, it was also evident that the observed cellulose hydrolysis is tightly coupled to an active microbial population, since significant net hydrolysis only occurred in frozen samples where microorganisms remained viable. Thus, the inherent enzymatic capacity originally present in the soils samples is lost when soils freeze, either because of the direct influence of temperature on

enzyme stability²⁷ or an indirect effect on enzyme mobility due to reduced diffusion rates in the frozen matrix¹³. However, it is evident that the microbial activity at -4°C led to the production of new functional enzymes capable of cellulose hydrolysis in the frozen soil”.

Line 301: Do you mean enrichment in CO₂?

Response and action : No. Here we mean ¹³C enrichment of organic molecular fractions. However, after the revision this sentence has been deleted from the manuscript.

Line 302: I think “microbial activity” is more suitable than “microbial growth rates”.

Action: Changed accordingly.

Lines 314-316: I cannot follow this sentence. What do you mean with “temporal partitioning in utilization of the polymer substrate by different microbial groups”?

Response and action: We originally meant that different microbial groups were involved in metabolically utilizing and growing on the added substrate during different parts of the incubation. However, after the revision of data interpretation (see responses to reviewer#2 comments above), this sentence has been deleted from the manuscript.

Lines 323-326: None of the three cited references shows an increasing importance of fungi from early to late winter! Please check your references.

Response and action: The interpretation of the data relating to the end of the incubations (days 113-195) has now been deleted from the manuscript acknowledging reviewer’s #2 concerns on the evaluation of the data given the few data points available and thus this sentence has been removed in the revised manuscript.

Lines 327-329: The concentration of WSC is determined by the balance between production and consumption. A decrease in WSC could result from a decrease in cellulose breakdown, but also from an increased uptake of WSC by microorganisms (at constant, reduced or increased cellulose breakdown rates).

Response and action: We agree with the reviewer that the ¹³C-WSCs we observe is the net value and its abundance is influenced by both production and consumption which may to a large degree explain the large variations observed in ¹³C-WSCs (figure 4b).

In addition to the revisions of the figure 4b outlined in page 10 above, we have added information on the discussion (lines 163-172).

Lines 329-336: This is very speculative, and I would like to see it phrased more carefully.

Response: This concern is dealt with in the substantial revisions in data evaluation and discussion in response to all reviewers’ comments.

Action: We have changed this text in lines (187-196) in the discussion.

Figures

Figure 2. I do not find it necessary to plot the different markers on different scales (Figure 2b). The figure would be easier to read if all were on the same scale. Where the logarithmic functions significant?

Action: We have removed the 2nd y-axis in panel b to enhance clarity according to the reviewer's request. The function in Figure 2b has been changed to a linear function adopting the same scheme as applied for the CO₂ and ¹³C WSC data in Fig 1a-b and 4 a-b with a linear increase up to 113 days. In this regression we have, however, excluded the time point at zero days, since it is evident that the amounts of PLFAs drop during the initial part of the incubation. We motivate this with the fact that this kind of initial microbial response to freezing has been documented in the literature (e.g. Feng *et al.*, 2007; Schmitt *et al.*, 2008). The logarithmic functions proposed in the original manuscript were significant (p<0.05).

Figure 3. Change “cell membrane lipids” to “phospholipid fatty acids” in the legend, unless the NMR method was not specific for PLFAs.

Action: Changed accordingly.

Figure 5. Change the x axes labels to “bacterial and actinobacterial PLFAs” (Figure 5a) and “fungal PLFAs” (Figure 5b).

Action: Figures are removed from the revised manuscript; see motivation in response to reviewer # 2 comments above.

Specific references referred to in the responses to reviewer's comments

- Ab Rani M a, Brant A, Crowhurst L et al. (2011) Understanding the polarity of ionic liquids. *Physical chemistry chemical physics : PCCP*, **13**, 16831–16840.
- Bax A, Morris GA (1981) An improved method for heteronuclear chemical shift correlation by two-dimensional NMR. *Journal of Magnetic Resonance (1969)*, **42**, 501–505.
- Clough MT, Geyer K, Hunt PA, Son S, Vagt U, Welton T (2015) Ionic liquids: not always innocent solvents for cellulose. *Green Chem.*, **17**, 231–243.
- Fan TWM, Bird JA, Brodie EL, Lane AN (2009) ¹³C-Isotopomer-based metabolomics of microbial groups isolated from two forest soils. *Metabolomics*, **5**, 108–122.
- Feng X, Nielsen LL, Simpson MJ (2007) Responses of soil organic matter and microorganisms to freeze–thaw cycles. *Soil Biology and Biochemistry*, **39**, 2027–2037.
- Frostegård A, Tunlid A, Baath E (1991) Microbial biomass measured as total lipid phosphate in soil of different organic content. *Journal of Microbiological Methods*, **14**, 151–163.
- Hansen SH, McCormack JG (2002) Application of ¹³C-filtered ¹H NMR to evaluate drug action on gluconeogenesis and glycogenolysis simultaneously in isolated rat hepatocytes. *NMR in Biomedicine*, **15**, 313–319.

- Mahboubi A, Linden P, Hedenström M, Moritz T, Niittylä T (2015) ^{13}C Tracking after $^{13}\text{CO}_2$ Supply Revealed Diurnal Patterns of Wood Formation in Aspen. *Plant Physiol.*, **168**, 478–489.
- Mahrous E a, Lee RB, Lee RE (2008) A rapid approach to lipid profiling of mycobacteria using 2D HSQC NMR maps. *Journal of lipid research*, **49**, 455–463.
- Martineau E, Akoka S, Boisseau R, Delanoue B, Giraudeau P (2013) Fast Quantitative ^1H – ^{13}C Two-Dimensional NMR with Very High Precision. *Analytical Chemistry*, **85**, 4777–4783.
- Morris GA (1980) Indirect measurement of proton relaxation rates by “INEPT” polarization transfer to carbon-13: Proton spin-lattice relaxation in cholesteryl acetate solutions. *Journal of Magnetic Resonance (1969)*, **41**, 185–188.
- Schmitt A, Glaser B, Borken W, Matzner E (2008) Repeated freeze-thaw cycles changed organic matter quality in a temperate forest soil. *Journal of Plant Nutrition and Soil Science*, **171**, 707–718.
- Szyperski T (1995) Biosynthetically Directed Fractional C-13-Labeling of Proteinogenic Amino-Acids - an Efficient Analytical Tool to Investigate Intermediary Metabolism. *European Journal of Biochemistry*, **232**, 433–448.
- Vogel JG, Valentine DW, Ruess RW (2005) Soil and root respiration in mature Alaskan black spruce forests that vary in soil organic matter decomposition rates. *Canadian Journal of Forest Research*, **35**, 161–174.
- Werner RA, Bruch BA, Brand WA (1999) ConFlo III—An Interface for High Precision d ^{13}C and d ^{15}N Analysis with an Extended Dynamic Range. *Rapid Commun. Mass Spectrom*, **13**, 1237–1241.

Reviewers' Comments:

Reviewer #1:

Remarks to the Author:

With the incorporation of the previously suggested edits, the authors have produced a timely, interesting, and experimentally sound contribution to the field.

There are a few minor points below:

Line 24: this sentence is confusing as written.

Line 221: "soil"

Line 247: this sentence is important but hard to understand as written, please clarify the wording

Reviewer #2:

Remarks to the Author:

I am satisfied with the authors' thorough and thoughtful responses to the points raised and recommend publication without further change.

Reviewer #4:

Remarks to the Author:

I find the manuscript much improved. The authors did a good job highlighting the impact and novelty of their research, and the reduction of the discussion helps to focus on the main message – the ongoing degradation of an important biopolymer in frozen soils. I have only a few small comments left.

Line 31: Typo in "conceptualizing". I'm also not sure this is the right word.

Line 168: Typo, change to "soil samples".

Line 186: "that" missing.

Lines 218-221: I find the sentence difficult to follow (but I understand what you mean). Please rephrase.

Line 248: Typo, Change to "soils".

Line 268: Change "is" to "are".

Line 306: I suppose you corrected for the CO₂ background when calculating CO₂ and ¹³CO₂ production rates. Can you add a comment on that?

Line 362: Spelling of actinobacterial markers is not consistent.

Response to Reviewer's comments (actions taken in *italic font*).

REVIEWERS' COMMENTS:

Reviewer #1 (Remarks to the Author):

With the incorporation of the previously suggested edits, the authors have produced a timely, interesting, and experimentally sound contribution to the field.

There are a few minor points below:

Line 24: this sentence is confusing as written.

Action: The section of the abstract where this sentence originally appeared has been revised according to an "editorial request". The corresponding information now reads "For winter SOM decomposition to have a substantial influence on soil carbon balances it is crucial whether or not biopolymers can be metabolized in frozen soils.", which should enhance clarity. (Lines 23 to 25)

Line 221: "soil"

Action: Changed accordingly

Line 247: this sentence is important but hard to understand as written, please clarify the wording

Action: The sentence has been revised and broken down into two sentences to enhance clarity. It now reads: "However, it has also been documented that the role of e.g. substrate and nutrient availability on microbial activity becomes less important when soils freeze¹⁷. When soils freeze most of the soil pore water transition into ice and the dissolved compounds become more concentrated in the remaining liquid water pool¹⁶, at the same time as metabolic rates and nutrient demands are decreased.". (Lines 254 to 257)

Reviewer #2 (Remarks to the Author):

I am satisfied with the authors' thorough and thoughtful responses to the points raised and recommend publication without further change.

Reviewer #4 (Remarks to the Author):

I find the manuscript much improved. The authors did a good job highlighting the impact and novelty of their research, and the reduction of the discussion helps to focus on the main message – the ongoing degradation of an important biopolymer in frozen soils. I have only a few small comments left.

Line 31: Typo in "conceptualizing". I'm also not sure this is the right word.

Action: "conceptualizing" changed to "understanding"

Line 168: Typo, change to "soil samples".

Action: Changed accordingly

Line 186: "that" missing.

Action: Changed accordingly

Lines 218-221: I find the sentence difficult to follow (but I understand what you mean). Please re-phrase.

*Action: For clarity, we have broken it down to two sentences and it now reads:
“Understanding of the contribution from winter and frozen conditions to the soil carbon balance is essential given the long winters in high latitude ecosystems. In addition, accounting for all contributions to SOM decomposition, and especially the biopolymeric constituents, is crucial.” (Lines 224 to 227)*

Line 248: Typo, Change to “soils”.

Action: Changed accordingly

Line 268: Change “is” to “are”.

Action: Changed accordingly

Line 306: I suppose you corrected for the CO₂ background when calculating CO₂ and ¹³CO₂ production rates. Can you add a comment on that?

Action: The background CO₂ was corrected for and we added the sentence “The background CO₂ and ¹³CO₂ concentrations were corrected for when calculating production rates.” to enhance clarity (Line 357 to 358)

Line 362: Spelling of actinobacterial markers is not consistent.

Action: The spelling of “actinobacterial markers” has been revised for consistency.